# Health Risk Assessment of Exposure to 15 Essential and Toxic Elements in Spanish Women of Reproductive Age: A Case Study

**DOI:** 10.3390/ijerph182413012

**Published:** 2021-12-09

**Authors:** Carmen Sáez, Alfredo Sánchez, Vicent Yusà, Pablo Dualde, Sandra F. Fernández, Antonio López, Francisca Corpas-Burgos, Miguel Ángel Aguirre, Clara Coscollà

**Affiliations:** 1Public Health Laboratory of Alicante, 6 Plaza de España, 03010 Alicante, Spain; saez_cartor@gva.es (C.S.); sanchez_alfagu@gva.es (A.S.); 2Department of Analytical Chemistry, Nutrition and Food Science, Institute of Materials, University of Alicante, 03080 Alicante, Spain; aguirre.pastor@ua.es; 3Foundation for the Promotion of Health and Biomedical Research in the Valencian Region, FISABIO-Public Health, 21, Avenida Catalunya, 46020 Valencia, Spain; yusa_vic@gva.es (V.Y.); dualde_pab@gva.es (P.D.); fernandez_sanfer@gva.es (S.F.F.); lopez_anttob@gva.es (A.L.); corpas_fra@gva.es (F.C.-B.); 4Public Health Laboratory of Valencia, 21, Avenida Catalunya, 46020 Valencia, Spain; 5Analytical Chemistry Department, University of Valencia, Edifici Jeroni Muñoz, Dr. Moliner 50, 46100 Burjassot, Spain

**Keywords:** biomonitoring, women, elements, urine, risk assessment

## Abstract

This case study investigates the exposure of 119 Spanish women of reproductive age to 5 essential (Co, Cu, Mn, V, Zn) and 10 toxic (Ba, Be, Cs, Ni, Pb, Pt, Sb, Th, Al, U) elements and assesses their risk. The essential elements (Co, Cu, Mn, V, and Zn) showed average concentrations (GM: geometric mean) of 0.8, 35, 0.5, 0.2, and 347 μg/L, respectively. Five of the toxic elements (Ba, Cs, Ni, Al, U) exhibited detection frequencies of 100%. The GM concentrations of the novel toxic elements were 12 μg/L (Al), 0.01 μg/L (Pt), 0.02 μg/L (U), 0.12 μg/L (Th), 0.009 μg/L (Be) and 4 μg/L (Cs). The urine analysis was combined with a survey to assess any variations between subgroups and potential predictors of exposure to elements in the female population. Significant differences were obtained between the rural and urban areas studied for the toxic element Cs, with higher levels found in mothers living in urban areas. In relation to diet, statistically significantly higher levels of essential (Cu) and toxic (Ba) elements were detected in women with a high consumption of fish, while mothers who consumed a large quantity of legumes presented higher levels of the toxic element Ni (*p* = 0.0134). In a risk-assessment context, hazard quotients (HQs) greater than 1 were only observed for the essential elements Zn and Cu in P95. No deficiency was found regarding the only essential element for which a biomonitoring equivalent for nutritional deficit is available (Zn). For the less-studied toxic elements (Al, Pt, U, Th, Be, and Cs), HQs were lower than 1, and thus, the health risk due to exposure to these elements is expected to be low for the female population under study.

## 1. Introduction

Reliable information about exposure to elements is of crucial importance for risk assessment and management. Yet, although the literature has many publications on the urinary levels of certain toxic elements in the female population, such as arsenic (As), cadmium (Cd), lead (Pb), mercury (Hg), antimony (Sb), barium (Ba), and nickel (Ni) [1,2,3,4,5,6,7,8,9,10,11,12,13,14,15,16,17,18,19,20], information on other toxic elements is scarce, as is the case of aluminum (Al), platinum (Pt), uranium (U), thorium (Th), beryllium (Be), and cesium (Cs).

Al is ubiquitous in the environment. For the general population, exposure to it most likely occurs through the consumption of food (mainly processed foods), water, and aluminum-containing medications, such as antacids, buffered analgesics, antidiarrheal, or antiulcer drugs. Urinary Al has been described as the best indicator of exposure, reflecting mainly current intake. In addition, Al has been found to be toxic for reproductive processes [21].

Pt belongs to the platinum group elements (PGEs). The PGEs in catalysts contribute to the decrease of pollutants in exhaust gases. However, the increased use of PGEs as components of autocatalytic converters attached to motor vehicles has resulted in the urban environment being seriously polluted by Pt. Several studies have suggested that PGEs and their complexes in airborne particulate matter (PM) may be one of the causes of health disorders such as asthma, allergies, and rhinoconjunctivitis [22,23].

U is a primordial radioactive element that is omnipresent in the earth’s crust. It is found in all components of the environment and reaches the human body mostly through ingestion of drinking water and foodstuffs. U and its compounds are highly toxic both from a chemical and radiological standpoint [24].

Th is used to make ceramics and lantern mantles and can also be found in elements employed in the aerospace industry and in nuclear reactions. Additionally, Th can serve as a fuel for generating nuclear energy, and more than 30 years ago, thorium oxides were used in hospitals for certain kinds of diagnostic X-rays. Studies carried out on workers exposed to Th have shown that breathing thorium dust may cause an increased chance of developing lung disease and lung or pancreas cancer many years after exposure [25].

Be compounds are commercially mined and used in mirrors and special alloys for the automobile, computer, nuclear, energy, aircraft, and machine-parts industries. This element is also utilized in the production of sports equipment such as golf clubs and bike frames, while in medicine, it can be found in instruments, X-ray machines, and dental bridges. Exposure to Be occurs primarily in the workplace, near certain hazardous waste sites, and from breathing tobacco smoke. The general population can be exposed to low levels of Be through breathing air, eating food, or drinking water containing this element. Air levels of Be higher than 100 μg/m^3^ can result in erythema and edema of the lung mucosa, which leads to pneumonitis [26].

Cs is a silver-white element that is found naturally in rock, soil, and clay. Inorganic cesium compounds are used in photomultiplier and vacuum tubes, scintillation counters, infrared lamps, semiconductors, photographic emulsions, high-power gas-ion devices, and as polymerization catalysts. Radioactive 137Cs has been used medically to treat cancer. Regarding human exposure to Cs, most of it occurs through diet. For absorbed cesium salts, the biological half-life is estimated to be 70–109 days. Case investigations where large doses of cesium chloride were ingested have reported decreased appetite, nausea, diarrhea, and cardiac arrhythmia [27].

Zinc (Zn), copper (Cu), cobalt (Co), manganese (Mn), and vanadium (V) are essential elements that play an important role in the growth, development, and maintenance of the physiological homeostasis of organisms. Moreover, given that an excess or deficiency of these trace elements has harmful impacts on human health [28], it is crucial to ensure an adequate intake to prevent deficiencies [29]. In contrast, their accumulation over a long period of time in some systems of the human body, such as the central nervous, gastrointestinal, cardiovascular, hematopoietic, and renal systems, can pose serious health problems [30,31,32].

The presence of toxic metals/elements in food at high levels can have potentially harmful implications for the population [33,34]. The nature and severity of element toxicity depend on several factors: exposure level, chemical species (inorganic versus organic), route and mode of exposure (acute versus chronic), and gender and age of the exposed individual [32,35]. Women of reproductive age and young children are particularly vulnerable to element exposure because of their high nutritional needs and, thus, their increased gastrointestinal absorption of these elements [31].

Element exposure can be assessed through both external (food, water…) [33] and internal (biomonitoring in blood, urine, hair…) [2,36] approaches. The biomonitoring of trace elements in urine is considered a suitable and sensitive approach to determine human exposure to them. Some authors have studied the levels of elements in pregnant women’s urine as it is a noninvasive matrix that can be used when assessing maternal exposure to harmful substances [7,9,10,12,13,14,37] and allows the implementation of risk assessment studies. In previous research, we studied the exposure and risk of mothers of reproductive age in the Valencia province (Spain) to classical elements for which risk-based benchmark values have been established, such as Hg, Cd, As, Thallium (Tl), selenium (Se), and molybdenum (Mo) [1].

The present case study aims to complement the one mentioned above, assessing the exposure of women of reproductive age in the Valencian Community (Spain) to 5 essential elements (Cu, Co, Mn, V, and Zn) and 10 toxic elements (Ba, Be, Cs, Ni, Pb, Pt, Sb, Th, U, and Al) in relation to specific sources of exposure. We obtained information on the personal characteristics, diet, and lifestyle of the participating mothers and examined the association of urinary elements with physical, socio-demographic, and diet variables. Metal levels and determinants of exposure were compared with national and international studies.

To our knowledge, no research has been published yet on the levels of Pt in women’s urine, and moreover, no risk assessment has been carried out in human biomonitoring studies for more novel toxic elements, such as Al, Pt, U, Th, Be, and Cs.

## 2. Material and Methods

### 2.1. Study Area and Population

The present case study was performed at the University and Polytechnic Hospital “La Fe” (Valencia, Spain). Details of the study design have been described by Yusà et al. [38] and Coscollá et al. [1]. In short, 119 out of the 180 recruited women donated urine samples between June and December 2015. The mothers were aged from 20 to 45 years old. Characteristics of the studied population have been previously published by Coscollà et al. [1].

### 2.2. Samples and Data Collection

Urine sample collection was conducted between 2 and 8 weeks after birth. The mothers who agreed to participate signed the consent form and received a questionnaire in the hospital to complete at home. Additionally, they were provided with a kit (disposable gloves and a sterile 100 mL polypropylene bottle) and instructions on how to collect the first morning spot urine sample. They were informed about which day the questionnaire had to be filled in and when the urine sample should be collected.

The donors’ samples and data included in this study were managed by the IBSP-CV BioBank (PT13/0010/0064), which is part of the Spanish National Biobank Network and the Valencian Biobanking Network, and were processed following standard operating procedures with the approval of the Ethical and Scientific Committees. Samples were stored until analysis at −80 °C.

Self-reporting questionnaires with detailed information on socio-demographic characteristics, lifestyles, and diet (food consumption frequency) were administered to the studied population. Diet predictors of urinary metal concentrations were grouped in two food questionnaires, a food consumption frequency questionnaire and a 72 h reminder questionnaire. Information on food consumption frequency by food groups was converted into semi-quantitative intake data, as explained in detail by Yusà et al. [38]. In addition, a 72 h reminder questionnaire was provided in a face-to-face interview with mothers between 2 and 8 weeks after birth to assess any recent exposure to metals. Mothers were asked about the number of portions consumed by food groups in the last 72 h.

### 2.3. Chemical Analysis

The chemical analysis of the elements was performed at the Public Health Laboratory of Alicante (Spain). Appendix A shows the Limit of Quantification (LoQ) of the elements identified. Creatinine analysis was carried out according to the kinetic method based on the Jaffé alkaline picrate reaction [39], using an automatic analyzer (Linear Kroma, Holliston, MA, USA). Despite being more susceptible to interference than enzymatic methods, the Jaffé method was selected to analyze urinary creatinine concentrations because it is considerably less expensive and, therefore, more suitable for routine laboratory work [40,41]. Additionally, to ensure the validity of this method, the laboratory participates in two German External Quality Assessment Scheme (G-EQUAS) intercomparison programs per year as an external quality control measure, showing satisfactory results in all of them.

#### 2.3.1. Target Elements

Out of 15 investigated elements, 5 were essential elements, such as Cu, Co, Mn, V, and Zn, and 10 were moderately or highly toxic elements, like Ba, Be, Cs, Ni, Pb, Pt, Sb, Th, U, and Al. However, these moderately or highly toxic elements might fulfill some essential functions at low levels [42].

#### 2.3.2. Element Determination by ICP-MS

The elements were studied on an ELAN DRC II ICP-MS PerkinElmer (Waltham, MA, USA), equipped with a perfluoroalkoxy (PFA) standard concentric nebulizer and a Peltier cooled (PC3) baffled glass cyclonic spray chamber. The ICP-MS instrumental operating conditions can be seen in Appendix A. The isotopes measured, the instrumental mode, and the DRC conditions for each element are also described in Appendix A.

One analytical method combines the determination of Co, Cu, Mn, Ni, and Zn in DRC-mode using methane (CH_4_) as a reaction gas together with the rest of the analytes (Al, Ba, Be, Cs, Pb, Pt, Sb, Th, and U) examined in standard mode. The second one uses the DRC’s ability to move some analytes (such as V) to a new analytical mass, away from the interference.

#### 2.3.3. Internal Quality Control (QC) and Method Performance

The methodology includes the use of quality control samples (QCS). We chose five different QCS to monitor the analytical sequence: initial calibration verification (ICV), initial calibration blank (ICB), reagent blanks, certified reference materials (CRM), continuous calibration verification (CCV), and internal standard signal monitoring. The limits of detection (LoD) and quantification (LoQ) are presented in Appendix A, while Appendix A shows the results for the reference materials.

### 2.4. Scientific Ethics Committee

The study and sampling protocol were approved by the Scientific Ethics Committee of the Valencian Research Centre for Public Health (FISABIO) of the Valencian Government (General Directorate of Public Health) and the Biomedical Scientific Ethics Committee of the University and Polytechnic Hospital La Fe. Before being included in the study, the potential participants were told about the aim and relevance of the research. All the participants signed an informed consent form approved by the Ethics Committee. To protect their privacy, all the collected personal data and biological samples were encoded and used only for research purposes.

### 2.5. Statistical Analysis

Statistical analysis was performed with R software (version 3.5.2). Metal levels in urine below detection limit were imputed following the maximum likelihood estimation method [43]. This method assumes that the data are distributed according to a certain parametric distribution and estimates the parameters of this distribution by maximizing the probability of obtaining the observed sample. A log-normal distribution was assumed for metal levels in urine.

First, we carried out a descriptive analysis of all the study variables. Qualitative variables were described by absolute frequencies and percentages. Quantitative variables were summarized by their median and range. Additionally, metal levels in urine were summarized by calculating the 25th, 50th, 75th, and 95th percentiles (P25, P50, P75 and P95), arithmetic (AM) and geometric means (GM), range and standard deviation.

Multiple linear regression models were built to assess the relationship between the concentration of each metal in urine and socio-demographic and dietary variables. The logarithmic transformation of metal levels in urine was used to attain the normality of the response variables. So that the magnitude of regression coefficients could be directly comparable, the numerical explanatory variables were centered and divided by twice their standard deviation, as proposed by Gelman [44]. These models were built following a forward or backward variable selection procedure, based on the Bayesian information criterion (BIC) [45], and a *p*-value < 0.05 was considered statistically significant. Confidence intervals at 95% significance were calculated for the model coefficients in order to assess the accuracy and uncertainty of the sample statistical estimates. Regression assumptions and influence of potential outliers were verified with residual plots.

### 2.6. Risk Assessment

We conducted a risk assessment to determine the study population’s exposure to those target elements showing detection frequencies higher than 40% in urine. Although urinary biomonitoring guidance values, such as biomonitoring equivalents (BEs), have been defined in the literature for certain elements [46,47,48], food intake guidance values were only available for others. In these cases, reverse dosimetry was used to estimate the daily intake of the element based on its urinary concentration through the following equations (Equations (1) and (2)) [49,50]:(1)$${\mathrm{EDI}}_{\mathrm{Vol}}\left(\frac{\mu g}{\mathrm{kg}\;\mathrm{bw}\cdot \mathrm{day}}\right)=\frac{C_{U}\left(\frac{\mu g}{L}\right)\times V_{24h}}{\mathrm{FUE}\times \mathrm{BW}\left(\mathrm{kg}\right)}$$
(2)$${\mathrm{EDI}}_{\mathrm{Cre}}\left(\frac{\mu g}{\mathrm{kg}\;\mathrm{bw}\cdot \mathrm{day}}\right)=\frac{C_{U}\left(\frac{\mu g}{g\;\mathrm{Cre}}\right)\times {\mathrm{Cre}}_{24h}}{\mathrm{FUE}\times \mathrm{BW}\left(\mathrm{kg}\right)}$$

EDI: estimated daily intake.C_U_: concentration of the element in urine (GM and P95).V_24h_: total urinary volume excreted within 24 h for adults (1.6 L·day^−1^) [51].Cre_24h_: total urinary creatinine excreted within 24 h (1.03 g·day^−1^) [52].F_UE_: urinary excretion factor of the element (see Table 1).BW: mean body weight of the participants (60 kg).

The hazard quotient (HQ) allows us to evaluate the risk of exposure by comparing it to a health-based guidance value. If a BE is established, urinary concentrations are compared to the corresponding BE, as in the case of Zn (439 μg/L), Ba (192 μg/L), and Al (57 μg/L). However, when BEs are not defined, daily intake guidance values are used as a reference, such as tolerable daily intake (TDI), reference dose (RfD), or tolerable upper intake level (UL), to compare the calculated EDIs. The HQ was estimated using the following equation (Equation (3)):(3)$$\mathrm{HQ}=\frac{\mathrm{EDI}\;}{\mathrm{GV}}\;\mathrm{or}\;\frac{C_{U}}{\mathrm{BE}}$$

HQ: hazard quotient.EDI: estimated daily intake.GV: oral intake health-based guidance values, such as tolerable daily intake (TDI), reference dose (RfD) or tolerable upper intake level (UL).C_U_: urinary concentration of the metal (at GM and P95 distribution levels).BE: biomonitoring equivalent.

For toxic and essential elements, exposure to the contaminant is not expected to be a health risk, if the HQ is lower than 1.

Different risk-based benchmarks were established for the essential elements. For instance, three reference values were set for Zn: a biomonitoring equivalent for the nutritional deficit related to estimated average requirement (BE_EAR_ = 159 μg/L), a biomonitoring equivalent (BE = 439 μg/L), and a biomonitoring equivalent point of departure (BE_POD_ = 1316 μg/L). In this study, the HQ for Zn was calculated using its BE (439 μg/L). Given that BEs are not defined for other essential elements, this approach cannot be used. This is the case of Co, Cu, and V, for which the calculated estimated daily intakes (EDIs) were compared to daily intake reference guidance values, such as reference dose (RfD) for Co (30 μg/kg/day), or tolerable upper intake level (UL) for Cu (5 mg/day) and V (1.8 mg/day) [53,54,55,56,57,58,59].

Regarding toxic elements, BEs have only been defined for two of them (Ba and Al), at 192 μg/L and 57 μg/L, respectively. The others were compared to daily intake reference guidance values, such as RfD (Sb = 0.4 μg/kg/day), TDI (Ni = 13 μg/kg/day and U = 0.6 μg/kg/day), BMDL_10_ (Pb = 0.63 μg/kg/day), and PDE: permitted daily exposure (Pt = 2.6 μg/kg/day) [60,61,62,63,64,65,66,67,68,69,70].

## 3. Results

### 3.1. Urinary Levels of Metals

Because urine is the main route of excretion of many elements, urinary levels have been used to reflect previous exposure to elements within a few hours to days [70]. Table 2 presents the results of the levels of elements detected in the urine of Spanish women of reproductive age as non-creatinine-adjusted and creatinine-adjusted concentrations.

Essential elements were found in all analyzed samples (100%), except for V, which was also detected at high frequency (79%). Additionally, non-essential or toxic elements (Ba, Cs, Ni, Al, and U) were identified in all samples (100%) or at high frequencies (Pb, Pt, Sb, and Th), ranging from 81 to 99%.

The novel and less-studied elements (Pt, U, Al, Th, Be, and Cs) presented geometric means of 0.01, 0.02, 12, 0.12, 0.009, and 4 μg/g creat, respectively. Figure 1 depicts the histograms for the concentrations of some elements.

### 3.2. Factors of Influence on the Urine Levels of Elements

Table 3 shows the results of the regression models obtained for each assessed element. The place of residence was relevant for the toxic element Cs, with mothers who had lived in rural locations for the last ten years presenting lower levels than those who lived in urban areas (*p* = 0.032).

From the different food groups investigated in the 72 h reminder questionnaire, consumption of fish, shellfish products, and legumes showed significant positive associations with the levels of essential (Cu: *p* = 0.0058) and toxic (Ba: *p* = 0.0153, Ni: *p* = 0.0134) elements detected in the urine. Concerning the rest of the assessed elements, significant associations were only found with creatinine levels.

### 3.3. Risk Assessment

Only EDIs for three essential (Co, Cu, and V) and seven toxic (Cs, Ni, Pb, Pt, Sb, Th, and U) elements at GM and P95 levels were calculated for their urinary concentrations, adjusted by volume and creatinine. EDIs for elements presenting DF < 40% (Be), unavailable BEs for essential (Zn) and toxic elements (Ba, Al) or missing F_UE_ values (Mn) were not estimated. Table 4 presents the resulting EDIs.

As can be seen in that table, the EDI based on creatinine for essential elements ranged from 0.4 (V) to 35 μg/kg/day (Cu) in the GM, and from 1.7 (V) to 64 μg/kg/day (Cu) in the 95th percentile. Regarding toxic elements, the lowest EDIs based on creatinine were found for Pt (0.0003 μg/kg/day) in the GM, and the highest EDI values were observed in the case of Ni (1.8 μg/kg/day).

The risk of exposure to elements was assessed by calculating HQ ratios based on EDIs for Co, Cu, V, Ni, Pb, Pt, Sb, and U, and based on BEs for Zn, Ba, and Al, adjusted by volume and creatinine. HQs were not calculated for Be (DF < 40%), Mn (lack of toxicokinetic data and BE), Cs and Th (lack of an established reference health-based value). Figure 2 shows the results of the risk assessment in the target population. According to this figure, HQs at the GM level were higher in Zn (0.62–0.79) and Cu (0.42–0.65), followed by Pb (0.18–0.28), Ni (0.14–0.23), and Al (0.22), but in all cases they were lower than 1. The lowest HQs were derived for Pt, ranging from 0.00013 (GM) to 0.0014 (P95). Only for Cu and Zn in the 95th percentile, HQs were higher than 1, since the obtained concentrations were above the BE and UL, respectively.

## 4. Discussion

### 4.1. Urinary Levels

In Table 5, we can see comparative data on the urinary levels of elements in women from other studies. Of the different essential elements, only Zn presented similar concentrations to those found in this study. Snoj Tratnik et al. [71] detected a P95 concentration of 824 μg/L for Zn in the Slovenian population, which is consistent with our P95 results (848.10 μg/L). In general, higher concentrations of essential elements (Co, Cu, Mn, and V) were identified in the Spanish women under study compared to other works (see Table 5). Specifically, GM concentrations of Cu in the women from this research were found to be higher than those observed in women living in Japan [11,19], Canada [16], and Belgium [17]. Furthermore, Snoj-Tratnik et al. [71] detected a lower P95 concentration (22.5 μg/L) in Slovenian women than in the Spanish women presented here (78.68 μg/L). With regard to Co, it is used in the formation of vitamin B12, and 85% of the human body’s content of this metal is in this form [29]. Its arithmetic mean (AM) in the studied population was again higher than the one reported by Avino et al. [2] for the Italian population. In the case of Mn, although it facilitates many enzymatic reactions, high levels of it have been associated with delays in children’s neurodevelopment. In addition, some prospective studies have discovered that high prenatal and early postnatal exposure to Mn may be associated with adverse effects on neurodevelopment [72]. Urinary Mn concentrations in the Spanish women participating in our research, American women [73], and Saudi women [18] were observed to be higher than in the American NANHES group [73] and Japanese women [19]. V is involved in insulin function, but a high exposure to it may be toxic to the human body, especially to pregnant women, since it can affect the fetus increasing the risk of multiple adverse birth outcomes, such as low birth weight, preterm birth, low gestational age, and premature rupture of membranes [74,75]. Moreover, Wang et al. [76] demonstrated that V levels were high in women who had suffered a miscarriage. In our study, the GM concentrations of V in urine were greater than those found in Bedouin Arab mothers [5], while the AM observed here was lower than those reported for the Chinese population [28].

Concerning the toxic elements that have been more widely studied, evidence shows that mothers exposed to metals such as Pb during pregnancy have a high risk of preterm births, spontaneous abortions, hypertension, low birth weight, and stillbirth [80]. According to our findings, Pb was detected in 99.1% of the maternal urine samples with GM = 0.795 μg/L, which is similar to the levels identified in mothers in the USA [9,10]. However, the women in our study had lower urinary Pb levels compared to pregnant women in China (AM = 6.35 μg/L and GM = 3.35 μg/L) [7], and Nigeria (AM = 28.5 μg/g creat) [14]. It is important to note that the problem these countries have with Pb contamination in the environment (water, food, and air contamination) [81,82,83] could account for the differences mentioned above. In another Spanish survey on pregnant women (N = 489) carried out in 2004–2006, the authors detected higher creatinine-adjusted concentrations of Pb in maternal urine (5.2 μg/g creat) during the third trimester of pregnancy than those found in our study (0.91 μg/g creat). This could be explained by predicted increases of Pb mobilization from the bone tissues during the stages of pregnancy [13]. Table 5 shows that the Sb concentrations in our study are similar to those reported for pregnant women in the USA [4] and France [3], but higher than those reported for the healthy Italian population [2]. Our geometric mean concentration of Ni was the highest compared to other European countries (Italy [2], France [20], and Belgium [17]), Japan [19], and Canada [16]). For Ba, the GM of Bedouin Arab mothers [5] was lower than the one detected here, whereas pregnant women in Puerto Rico and the USA presented higher concentrations than the Spanish women under study [4,6].

Among the less studied toxic elements, such as Pt, U, Al, Th, Be, and Cs, it should be highlighted that Pt in the urine of females has not been investigated before, and consequently, this is the first study to do so. For the other elements, there are a few works in the literature in which they have been analyzed. For instance, the median concentration of Cs in the mothers’ urine samples was 4.27 μg/L in our study, which was lower than the level observed in the urine samples from pregnant women in Australia (6.35 μg/L) [84], the USA (5.06 μg/L) [9], China (6.68 μg/L) [76], and another Spanish study (8.0 μg/L) [13]. Very few works have reported urinary U concentrations for women or even the general population [73]. The GM urinary U concentration for the assessed Spanish women of reproductive age (0.02 μg/g creat) was higher than for US women (0.006 μg/g creat). According to Hoover et al. [73], the median U concentration in urine for women in the Navajo Birth Cohort Study (NBCS) was 2.67–2.8 times higher than the U.S. reference population. Be and U were the elements with the lowest concentrations found in this study. The detection rate for Be was only 32%, which may be one of the reasons why few authors excluded it in their studies [4]. In fact, only Bedouin Arab mothers presented quantifiable levels of this element (GM = 0.11 μg/L) [5]. Meanwhile, the concentration of Th found in Spanish women was similar to that of Chinese women (AM = 0.17 μg/L) [79], and in the case of Al, the geometric mean of Spanish women was higher than Bedouin Arab mothers living in Israel [5].

### 4.2. Predictors of Exposure

Our study examined factors that may influence the urinary element levels in mothers. One of the mothers’ personal characteristics for which we found an association with element levels was the specific creatinine value measured for each individual. Considering that creatinine standardization may cause bias in the study results, we performed a sensitivity analysis introducing creatinine into the model as a covariate instead of using the creatinine-standardized element concentrations [85]. Multiple linear regression models showed a positive relationship between creatinine levels of mothers and all the studied elements except for Be, Pt, Th, and Al, for which no significant association was found. The National Health and Nutrition Examination Survey (NHANES) (years 2003 to 2010) in the U.S. also reported that the levels of the studied elements rose as the level of urine creatinine increased [9].

A positive association was observed between the mother’s place of residence and the occurrence of toxic elements in urine. In this sense, we found significantly lower Cs levels in the urine of women who lived in rural areas (GM = 3.66 μg/g creat) compared to mothers living in urban areas (GM = 4.29 μg/g creat). Factors related to rural living, such as less time spent in traffic per day and no potential environmental Cs contamination from industrial emissions, could explain our results.

In terms of dietary intake habits, women who consumed legumes and fish products more frequently had higher levels of essential (Cu) and toxic (Ni, Ba) elements in their urine. Recently, Schrenk et al. [62] reported that the highest mean Ni concentrations in food commodities corresponded to the food category “Legumes, nuts and oilseeds” and, albeit to a lesser extent, fish was also found to be a source of Ni in the diet, which is in line with our results.

### 4.3. Risk Assessment

Of the four essential elements assessed, only one (Zn) has a BE to compare with the urinary concentration of the element. The rest of the essential elements have either a UL (Cu, V) or an RfD (Co). In the target population, HQs higher than 1 were only observed in Cu and Zn exposure when the P95 was considered, showing the need to reduce exposure to these elements. A deficiency in essential elements in the human body can also produce harmful effects. In the case of Zn, no deficiency was found because all mothers presented urinary levels of Zn higher than the BE for deficiency (159 μg/L). Unfortunately, nutritional adequacy could not be calculated for the other essential elements due to the lack of existing BE_EAR_.

After excessive consumption of Zn, the bioavailability of Cu can be suppressed [86]. However, although the P95 HQ of Zn was higher than 1, lack of Cu in the studied population was not expected since the Cu levels were higher than in other countries (Table 5).

Of the seven toxic elements analyzed, only two (Ba and Al) have BEs. For the others their EDIs were compared to daily intake reference guidance values, such as RfD (Sb), TDI (Ni and U), BMDL_10_ (Pb), and PDE (Pt). Given that the HQ obtained for the toxic elements was lower than 1, the health risk due to exposure of these elements, including the novel ones, was considered low for the population under study.

## 5. Conclusions

In this study, we estimated Spanish women’s exposure to toxic and essential elements by measuring their concentration in urine. Four essential elements (Co, Cu, Mn, and Zn) and five non-essential or toxic ones (Ba, Cs, Ni, Al, and U) were detected in all analyzed samples. In general, concentrations of toxic elements were higher than in previous studies. Among the toxic elements, we have investigated six new toxic elements presenting concentrations of 12 μg/L (Al), 0.01 μg/L (Pt), 0.02 μg/L (U), 0.12 μg/L (Th), 0.009 μg/L (Be), and 4 μg/L (Cs).

A positive association was observed between living in an urban area and urinary levels of the toxic element Cs. Furthermore, mothers who consumed more fish and legume products presented higher levels of essential (Cu) and toxic (Ba, Ni) elements in their urine samples.

To our knowledge, no research on the risk assessment of Co, Cu, V, Ni, Pb, Pt, Sb, U, Zn, Ba and Al has been previously carried out in women using human biomonitoring. Concerning the risk assessment conducted in this study, only exposure to Cu and Zn might be considered a concern in the target population, despite not being toxic elements. No nutritional deficiency was found for the studied essential element Zn. Nonetheless, Cu and Zn should be monitored to control whether the HQs above 1 (when P95 is considered) could be a risk factor and produce health problems in the studied population.

In general, for a better risk assessment, more studies should be carried out to set urinary biomonitoring guidance values, such as biomonitoring equivalents (BEs) for all elements. Moreover, it could be interesting for other works to establish BE_EAR_ for essential elements to be able to properly assess the possible deficiencies in the population.

**Study limitations:** Maternal urinary element levels were only measured using the first morning spot urine samples between 2 and 8 weeks after birth, which may not accurately reflect the long-term exposure levels of elements. In addition, although urine is a frequently used as a non-invasive matrix, it is not the most reliable matrix for certain elements (e.g., Mn). The choice of the Jaffé reaction for the analysis of urinary creatinine levels could have influenced the obtained results, since it is more susceptible to interference than other methods. Participants who joined the study were recruited from one hospital that covered only a part of the Valencian territory. Likewise, the education level of participants was skewed toward university studies. Consequently, the sample was not representative, and the conclusions should not be extrapolated to the entire population.

## Figures and Tables

**Figure 1 ijerph-18-13012-f001:**
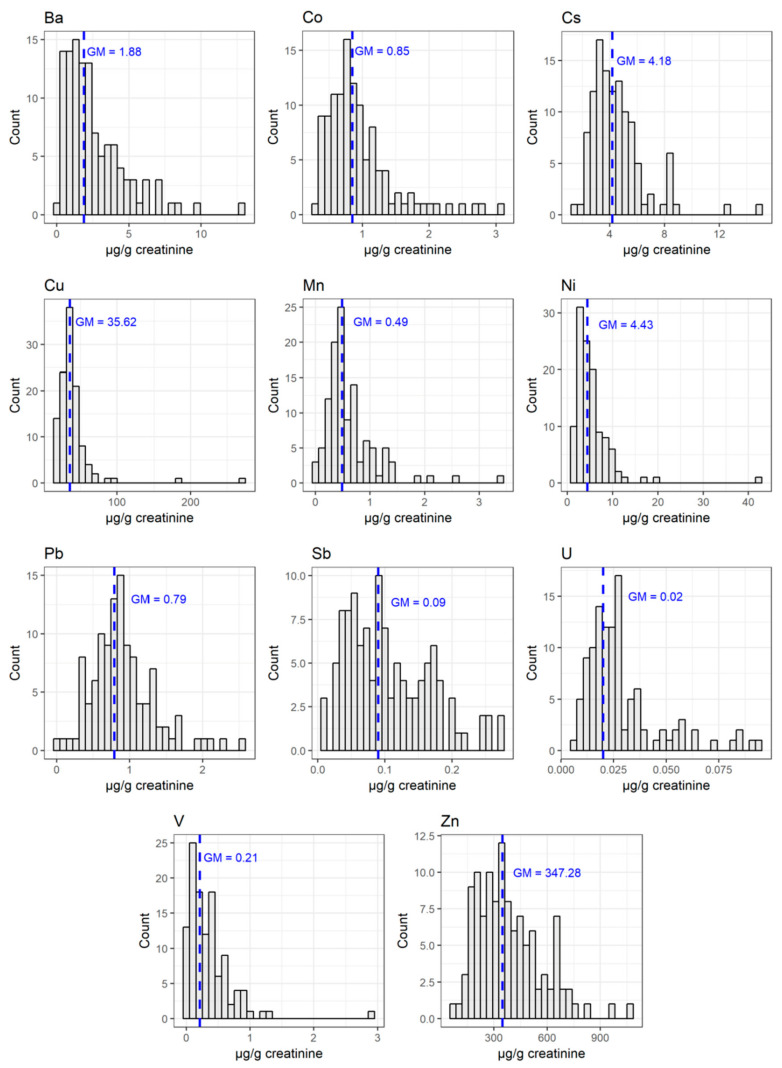
Histograms for some investigated elements (Ba, Co, Cs, Cu, Mn, Ni, Pb, Sb, V, Zn, and U). GM: geometric mean.

**Figure 2 ijerph-18-13012-f002:**
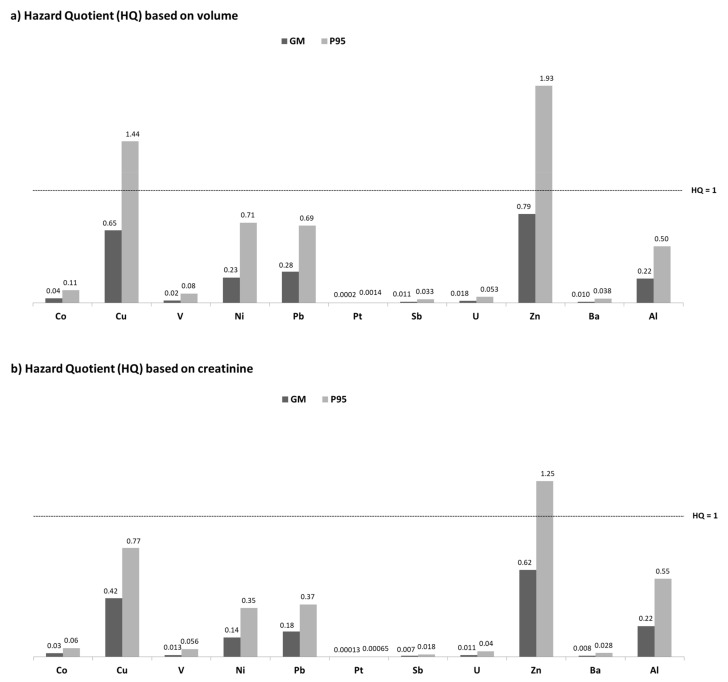
Hazard quotients for some metal concentrations based on volume (**a**) and creatinine (**b**). GM: geometric mean; P95: 95th percentile.

**Table 1 ijerph-18-13012-t001:** Urinary excretion factors and health-based reference values of the elements used for the risk assessment.

Metal Group	Element	F_UE_			Health-Based Reference Value				
		Value	Considerations	Ref.	Value	Type	Units	Considerations	Ref.
Essential metals	Co	0.019 (mean of 2 values)	Mostly excreted in feces after oral administration Data of cumulative urinary excretion in studies with animals following oral administration of Co_3_O_4_. After inhalation or dermal exposure Co is mainly excreted in urine.	[53]	30	RfD	μg/kg/day	POD/30	[54]
	Cu	0.018 (mean of 2 values)	Mostly excreted in bile after oral administration. Data of urinary excretion after daily intake. At high Cu intakes, urinary Cu excretion increases	[55]	5	UL (EFSA)	mg/day	NOAEL/2	[56]
	Mn	NF	Mn is mainly excreted in feces. Urinary excretion of Mn is not related with oral intake	[57]	NF	NF	NF	Owing to limitations of the human data and the non-availability of NOAELs, upper level cannot be set	[54]
	V	0.009	Study with rats. Data of 7-day cumulative excretion after oral administration as sodium metavanadate. Mostly excreted unabsorbed in feces after oral administration. After inhalation V is mainly excreted in urine.	[58]	1.8	UL	mg/day	UL for ≥19 years (LOAEL/300) Due to lack of data, it was not possible to determine ULs for pregnant and lactating women, children, and infants. These individuals should be particularly cautious about consuming vanadium supplements.	[59]
	Zn	NA	NA	NA	159 (204)	BE for deficiency	μg/L (μg/g)	Based on IOM EAR (nutritional) for women (0.1 mg/kg/day)	[46]
					439 (564) and 1316 (1693)	BE and BE_POD_ for toxicity	μg/L (μg/g)	ATSDR chronic MRL (0.3 mg/kg/day) ^1^	
Toxic metals ^2^	Ba	NA	NA	NA	192 (5.74)	BE (BE_POD_)	μg/L (mg/L)	Based on U.S. EPA’s RfD (0.2 mg/kg/day)	[47]
					246 (7.39)		μg/g (mg/g)		
	Cs	0.10 (mean of 2 values) ^3^	Data of 7-day cumulative excretion after oral administration as cesium chloride	[60]	NF	NF	NF	NF	NF
	Ni	0.04 (24 h)	Based on ingested dose of Ni chloride in rats	[61]	13	TDI	μg/kg/day	Oral exposure	[62]
	Pb	0.12 (24 h)	Based on ingested dose. Mostly excreted in feces.	[63]	0.63	BMDL_10_	μg/kg/day	Renal Effects, oral exposure	[43]
	Pt	0.51 (mean of 2 values, 24 h)	Approximation. Based on diet studies.	[63]	2.6	PDE	μg/kg/day	Oral exposure	[64]
	Sb	0.55 (mean of 2 values, 24 h)	Based on intravenous injection in humans	[65]	0.4	RfD	μg/kg/day	Hematologic, oral exposure	[66]
	Th	0.025 (24 h)	Mostly excreted in feces	[67]	NF	NF	NF	NF	NF
	Al	NA	NA	NA	57 (20500)	BE (BE_POD_)	μg/L, μg/g	Based on LOAEL (50 mg/kg/day)	[48]
	U	0.05 (24 h)	Based on ingested dose. Mostly excreted in feces.	[68]	0.6	TDI	μg/kg/day	Oral exposure	[69]

^1^ The lowest derived value was selected; ^2^ Be was not assessed due to its low DF (<40%); ^3^ Data derived from total excretion and urinary/fecal excretion ratios; Ref: reference; POD: point of departure; RfD: reference dose; UL: tolerable upper intake level; BMDL10: benchmark dose lower confidence limit, resulting in a 10% extra cancer incidence; PDE: permitted daily exposure; NA: not applicable; NF: not found.

**Table 2 ijerph-18-13012-t002:** Frequency of detection and levels of elements (μg/L) in the urine of women from the Valencian Region (N = 119) ^1^. Creatinine-adjusted urinary levels (μg/g creat) ^1^.

Element	DF %	Minimum	P25	Median	AM	GM	P75	P95	Maximum	SD
Al	100	2.9 (2.2)	9 (8)	12 (12)	15(16)	12 (12)	15 (17)	29 (32)	164 (139)	17 (18)
Ba	100	0.13 (0.09)	1.2 (1.1)	2.0 (1.9)	2.7 (2.6)	1.9 (1.9)	3.0 (3.5)	7.3 (7.0)	17.2 (12.9)	2.7 (2.2)
Be	32	0.002 (0.002)	0.006 (0.005)	0.008 (0.009)	0.01 (0.01)	0.009 (0.009)	0.01 (0.01)	0.02 (0.03)	0.03 (0.04)	0.007 (0.009)
Co	100	0.2 (0.3)	0.6 (0.6)	0.8 (0.8)	1.0 (1.0)	0.8 (0.9)	1.2 (1.1)	2.4 (2.1)	4.7 (3.1)	0.8 (0.5)
Cs	100	1.1 (1.6)	3 (3)	4 (4)	5 (4)	4 (4)	6 (5)	10 (8)	15 (15)	3 (2)
Cu	100	9 (15)	24 (28)	35 (35)	43 (40)	35 (35)	53 (41)	79 (66)	400 (268)	42 (28)
Mn	100	0.02 (0.04)	0.4 (0.3)	0.5 (0.5)	0.6 (0.6)	0.5 (0.5)	0.7 (0.7)	1.4 (1.4)	2.4 (3.4)	0.4 (0.5)
Ni	100	0.8 (1.0)	3 (3)	4 (4)	6 (5)	4 (4)	6 (6)	14 (10)	38 (42)	5 (5)
Pb	99	0.02 (0.02)	0.5 (0.6)	0.8 (0.9)	1.0 (0.9)	0.8 (0.8)	1.3 (1.1)	1.9 (1.6)	3.2 (2.6)	0.6 (0.4)
Pt	81	0.001 (0.001)	0.007 (0.005)	0.01 (0.01)	0.02 (0.02)	0.01 (0.01)	0.02 (0.02)	0.07 (0.05)	0.21 (0.11)	0.02 (0.02)
Sb	97	0.007 (0.009)	0.05 (0.05)	0.10 (0.09)	0.12 (0.11)	0.09 (0.09)	0.16 (0.15)	0.27 (0.23)	0.55 (0.29)	0.09 (0.06)
Th	98	0.01 (0.02)	0.04 (0.04)	0.14 (0.13)	0.17 (0.19)	0.12 (0.12)	0.23 (0.26)	0.43 (0.54)	1.12 (0.80)	0.16 (0.18)
U	100	0.005 (0.006)	0.02 (0.02)	0.02 (0.02)	0.03 (0.03)	0.02 (0.02)	0.03 (0.03)	0.06 (0.07)	0.13 (0.09)	0.02 (0.02)
V	79	0.004 (0.004)	0.09(0.10)	0.3 (0.3)	0.4 (0.3)	0.2 (0.2)	0.5 (0.4)	0.84 (0.89)	1.92 (2.91)	0.31 (0.36)
Zn	100	71 (80)	230 (250)	343 (353)	417 (389)	347 (348)	560 (491)	848 (705)	1526 (1078)	259 (184)

^1^ Left censored data (˂LoQ) were imputed, according to the maximum likelihood estimation method (See Section 2.5). DF: frequency of detection; AM: arithmetic mean; GM: geometric mean; SD: standard deviation.

**Table 3 ijerph-18-13012-t003:** Results of the multiple linear regression models for log (elements) concentrations in the urine of Spanish women.

Metals	Factors	Estimated Coefficients (95% CI)	Standard Error	*p*-Value ^1^
Cs	Place of residence (last 10 years):			
	Urban	-	-	-
	Rural	−0.1913 (−0.3659–−0.0168)	0.088	0.032
	Creatinine (mg/dL)	0.9526 (0.7998–1.1055)	0.0771	<0.001
Ni	Legumes (nº portions 72 h)	0.1586 (0.0335–0.2836)	0.0631	0.013
	Creatinine (mg/dL)	0.6844 (0.4693–0.8996)	0.1085	<0.001
Cu	Fish (nº portions 72 h)	0.146 (0.0433–0.2488)	0.0518	0.006
	Creatinine (mg/dL)	0.8887 (0.7246–1.0529)	0.0828	<0.001
Ba	Shellfish (nº portions 72 h)	0.355 (0.0695–0.6405)	0.144	0.015
	Creatinine (mg/dL)	0.6113 (0.2779–0.9447)	0.1682	<0.001
Co	Creatinine (mg/dL)	0.8458 (0.6641–1.0276)	0.0917	<0.001
Mn	Creatinine (mg/dL)	0.3297 (0.0889–0.5706)	0.1215	0.008
Pb	Creatinine (mg/dL)	0.9146 (0.6654–1.1638)	0.1257	<0.001
Sb	Creatinine (mg/dL)	0.9212 (0.6642–1.1782)	0.1297	<0.001
V	Creatinine (mg/dL)	0.5437 (0.1082–0.9792)	0.2197	0.015
Zn	Creatinine (mg/dL)	0.7762 (0.5853–0.9672)	0.0963	<0.001
U	Creatinine (mg/dL)	0.4267 (0.2349–0.6186)	0.0968	<0.001

^1^*p*-value ≤ 0.05.

**Table 4 ijerph-18-13012-t004:** Estimated daily intake for some of the assessed elements.

Group	Metal	EDI Based on Volume (μg/kg/day)		EDI Based on Creatinine (μg/kg/day)	
		EDI_GM_	EDI_P95_	EDI_GM_	EDI_P95_
Essential metals ^1^	Co	1.2	3.4	0.77	1.87
	Cu	54	120	35	64
	V	0.6	2.5	0.4	1.7
Toxic metals ^2^	Cs	1.1	2.6	0.68	1.69
	Ni	2.9	9.3	1.8	4.5
	Pb	0.18	0.43	0.11	0.23
	Pt	0.0005	0.0037	0.0003	0.0017
	Sb	0.004	0.013	0.003	0.007
	Th	0.13	0.46	0.08	0.29
	U	0.011	0.032	0.007	0.024

EDIs for Zn, Ba and Al were not calculated since we used the BE concept for risk assessment. ^1^ Mn was not assessed due to the lack of information about its excretion urinary factor. ^2^ Be was not assessed due to its low DF (<40%).

**Table 5 ijerph-18-13012-t005:** Comparative data on levels of metals in women’s urine in other studies.

	Study	Country (Location)	Year	Population/ Subjects	N	Age (Years)	AM	GM
Essential Elements								
**Cu**								
	Present study	Spain (Valencia)	2015	Women	119	20–45	43.67 μg/L 39.60 μg/g creat	35.36 μg/L 35.47 μg/g creat
	[19]	Japan (11 prefectures)	2000–2005	Women (n.o.)	1000	20–81	-	13.4 μg/L
	[11]	Japan	2007–2008	Pregnant women	78	-		12.8 μg/g creat
	[16]	Canada (Quebec)	2009–2011	Women	2992	6–79	-	10.0 μg/L
	[17]	Belgium	2010–2011	general population	1022	>18		6.94 μg/L
**Co**								
	Present study	Spain (Valencia)	2015	Women	119	20–45	1.04 μg/L 0.95 μg/g creat	0.84 μg/L 0.85 μg/g creat
	[2]	Italy (urban areas)	-	Healthy subjects	44	-	0.0009 μg/L	-
**Mn**								
	Present study	Spain (Valencia)	2015	Women	119	20–45	0.60 μg/L 0.62 μg/g creat	0.49 μg/L 0.49 μg/g creat
	[77]	Pakistan	1999–2005	Women	166	45–75	1.55 μg/L	-
	[19]	Japan (11 prefectures)	2000–2005	Women (n.o.)	1000	20–81	-	0.14 μg/L
	[73]	NHANES (USA)	2011–2016	Women	-	14–45	-	0.14 μg/g creat
	[73]	New Mexico (USA)	2010	Pregnant women	448	14–45	-	0.30 μg/g creat
	[18]	Riyadh (Saudi Arabia)	2011–2013	Lactating women	198	19–45	-	2.42 μg/L
**Zn**								
	Present study	Spain (Valencia)	2015	Women	119	20–45	417.29 μg/L 388.74 μg/g creat	347.25 μg/L 348.30 μg/g creat
	[2]	Italy (urban areas)	-	Healthy subjects	44	-	0.0012 μg/L	-
	[17]	Belgium	2010–2011	General population	1022	>18	-	227 μg/L
	[11]	Japan	2007–2008	Pregnant women	78	-		393 μg/g creat
	[16]	Canada (Quebec)	2009–2011	women	2992	6–79	-	250 μg/L
**V**								
	Present study	Spain (Valencia)	2015	Women	119	20–45	0.35 μg/L 0.34 μg/g creat	0.20 μg/L 0.21 μg/g creat
	[5]	Israel	2011–2013	Bedouin Arab mothers	111	18–41	-	0.04 μg/L
	[74]	China	2014–2016	Pregnant women	1865	24–35	0.77 μg/L 0.79 μg/g creat	-
	[3]	France	2011	Pregnant women	990	>18	-	0.28 μg/L
**Non-essential elements**								
**Al**								
	Present study	Spain (Valencia)	2015	Women	119	20–45	15.09 μg/L 16.30 μg/g creat	12.36 μg/L 12.47 μg/g creat
	[78]	Israel	2013		140	>18	-	7.6 μg/L
**Sb**								
	Present study	Spain (Valencia)	2015	Women	119	20–45	0.12 μg/L 0.10 μg/g creat	0.09 μg/L 0.08 μg/g creat
	[2]	Italy (urban areas)	-	Healthy subjects	44	-	0.003 μg/L	-
	[3]	France	2011	Pregnant women	990	>18	-	0.04 μg/L
	[4]	NHANES (USA)	1999–2016	Pregnant women	404	15–44	-	0.079 μg/L 0.078 μg/g creat
**Ba**								
	Present study	Spain (Valencia)	2015	Women	119	20–45	2.70 μg/L 2.61 μg/g creat	1.87 μg/L 1.87 μg/g creat
	[5]	Israel	2011–2013	Bedouin Arab mothers	111	18–41	-	1.29 μg/L
	[6]	Puerto Rico	2011–2017	Pregnant women	1285	18–40	-	2.5 μg/L
	[4]	NHANES (USA)	1999–2016	Pregnant women	404	15–44	-	2.01 μg/L 2.00 μg/g creat
**Be**								
	Present study	Spain (Valencia)	2015	Women	119	20–45	0.01 μg/L 0.01 μg/g creat	<LoQ
	[5]	Israel	2011–2013	Bedouin Arab mothers	111	18–41	-	0.11 μg/L
**Cs**								
	Present study	Spain (Valencia)	2015	Women	119	20–45	4.88 μg/L 4.51 μg/g creat	4.16 μg/L 4.17 μg/g creat
	[2]	Italy (urban areas)	-	healthy subjects	44	-	0.0054 μg/L	-
**Pb**								
	Present study	Spain (Valencia)	2015	Women	119	20–45	0.97 μg/L 0.91 μg/g creat	0.79 μg/L 0.79 μg/g creat
	[7]	China	2012–2014	Pregnant women	800	-	6.35 μg/L 13.67 μg/g creat	3.35 μg/L 7.99 μg/g creat
	[8]	China	2009–2010	General population	1647	6–60	-	2.85 μg/L
	[9]	USA	2003–2010	Pregnant women	1565	17–39	-	0.63 μg/L
	[10]	USA	2003–2004	Pregnant women	268	15–44	-	0.81 μg/L
	[11]	Japan	2007–2008	Pregnant women	78	-	1.19 μg/g creat	0.48 μg/g creat
	[12]	Australia	2008–2010	Pregnant women	157	19–44	0.66 μg/L 13.67 μg/g creat	-
	[13]	Spain (Catalonia)	2004–2006	Pregnant women	489	>16	5.2 μg/g creat	-
	[14]	Nigeria	2006–2008	Pregnant women	214	17–49	28.5 μg/g creat	-
	[15]	USA	2011–2012	Women	1242	>20	-	0.316 μg/L 0.430 μg/g creat
	[16]	Canada (Quebec)	2009–2011	Adults	5738	6–79	-	0.52 μg/L
	[17]	Belgium	2010–2011	Adults (n.o.)	1022	>18	0.74 μg/L	15.4 μg/L
	[18]	Riyadh (Saudi Arabia)	2011–2013	Lactating women	204	19–45	-	5.04 μg/L
**Ni**								
	Present study	Spain (Valencia)	2015	Women	119	20–45	5.52 μg/L 5.38 μg/g creat	4.41 μg/L 4.42 μg/g creat
	[2]	Italy (urban areas)	-	healthy subjects	44	-	0.047 μg/L	
	[19]	Japan (11 prefectures)	2000–2005	Women (n.o)	1000	20–81	-	2.1 μg/L
	[16]	Canada (Quebec)	2009–2011	Adults	2992	6–79	-	1.30 μg/L
	[20]	France	2006–2007	Adults	2000	18–74	-	1.23 μg/L
	[17]	Belgium	2010–2011	Adults (n.o)	1022	>18	-	1.73 μg/L
**Pt**								
	Present study	Spain (Valencia)	2015	Women	119	20–45	0.02 μg/L 0.02 μg/g creat	0.01 μg/L 0.01 μg/g creat
**Th**								
	Present study	Spain (Valencia)	2015	Women	119	20–45	0.17 μg/L 0.19 μg/g creat	0.12 μg/L 0.12 μg/g creat
	[79]	China	2014–2015	Pregnant women	598	>20	0.17 μg/L	
**U**								
	Present study	Spain (Valencia)	2015	Women	119	20–45	0.03 μg/L 0.03 μg/g creat	0.02 μg/L 0.02 μg/g creat
	[73]	NHANES (USA)	2011–2016	Women	-	14–45	-	0.006 μg/g creat
	[73]	New Mexico (USA)	2010	Pregnant women	449	14–45	-	0.016 μg/g creat

AM: arithmetic mean; GM: geometric mean; n.o.: non-occupationally.

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
