# Peer review of "Health Risk Assessment of Exposure to 15 Essential and Toxic Elements in Spanish Women of Reproductive Age: A Case Study"

_ijerph, 2021, doi:10.3390/ijerph182413012_

Round 1

Reviewer 1 Report

Line 146-148 and “Study limitations” line 462: the authors should justify the choice of the Jaffè alkaline picrate reaction method over the enzymatic one to determine creatinine and therefore evaluate the excretion of essential and toxic elements. The Jaffé method is vulnerable to plasma protein interference an important interindividual variable, so it is possible that the reported results can be influenced up to exceeding the p-value.

Reviewer 2 Report

-The introduction missed the part related to the exitance of toxic metals/elements in the meals. I would recommend the authors to consider explaining this fact at first and then turn to the importance of calculating hazard index based on urinary measurements of metals.  for this you may consider some recent texts as such: 

https://doi.org/10.1007/s11356-019-07477-w 

https://doi.org/10.1016/j.chemosphere.2019.125639

-The authors calculated a hazard quotient for essential elements, this part required a better elaboration. the references used are very superficial and the discussion is as well. I would recommend the authors to discuss the elemental balance in the body, that is when one essential element goes up the other will be down. 

This kind of study is not that much novel as claimed, therefore I would recommend the authors to introduce the concept of case study both in the title and in the sprite of the paper. at the end of the article relationship of this finding with similar contexts could be discussed. 

Round 2

Reviewer 2 Report

The authors addressed all comments in an appropriate manner. I would recommend the submission for publication. 

please only pay attention to the presence of typos in the text. and please add the complete form of abbreviations also in figure 2 legend and table 5 caption. 
